# Chondrocyte Thrombomodulin Protects against Osteoarthritis

**DOI:** 10.3390/ijms24119522

**Published:** 2023-05-30

**Authors:** Lin Kang, Ai-Lun Yang, Chao-Han Lai, Tsan-Ju Chen, Sung-Yen Lin, Yan-Hsiung Wang, Chau-Zen Wang, Edward M. Conway, Hua-Lin Wu, Mei-Ling Ho, Je-Ken Chang, Chung-Hwan Chen, Tsung-Lin Cheng

**Affiliations:** 1Department of Obstetrics and Gynecology, National Cheng Kung University Hospital, College of Medicine, National Cheng Kung University, Tainan 704302, Taiwan; 2Institute of Sports Sciences, University of Taipei, Taipei 11153, Taiwan; 3Department of Biochemistry and Molecular Biology, College of Medicine, National Cheng Kung University, Tainan 701401, Taiwan; 4Department of Surgery, National Cheng Kung University Hospital, College of Medicine, National Cheng Kung University, Tainan 704302, Taiwan; 5Department of Biostatistics, Vanderbilt University Medical Center, Nashville, TN 37232, USA; 6Department of Physiology, School of Medicine, College of Medicine, Kaohsiung Medical University, Kaohsiung 807, Taiwan; 7Department of Orthopaedic Surgery, Kaohsiung Medical University Hospital, Kaohsiung Medical University, Kaohsiung 807, Taiwan; 8Departments of Orthopaedics, School of Medicine, College of Medicine, Kaohsiung Medical University, Kaohsiung 807, Taiwan; 9Orthopaedic Research Center, Kaohsiung Medical University, Kaohsiung 807, Taiwan; 10Regeneration Medicine and Cell Therapy Research Center, Kaohsiung Medical University, Kaohsiung 807, Taiwan; 11School of Dentistry, College of Dental Medicine, Kaohsiung Medical University, Kaohsiung 807, Taiwan; 12Graduate Institute of Medicine, College of Medicine, Kaohsiung Medical University, Kaohsiung 807, Taiwan; 13Department of Medical Research, Kaohsiung Medical University Hospital, Kaohsiung 807, Taiwan; 14College of Professional Studies, National Pingtung University of Science and Technology, Pingtung 912, Taiwan; 15Centre for Blood Research, Faculty of Medicine, Life Sciences Institute, University of British Columbia, Vancouver, BC V6T 1Z3, Canada; 16Department of Orthopedics, Kaohsiung Municipal Ta-Tung Hospital, Kaohsiung 800, Taiwan; 17Department of Marine Biotechnology and Resources, National Sun Yat-sen University, Kaohsiung 80424, Taiwan

**Keywords:** thrombomodulin, osteoarthritis, miRNA, transgenic mice

## Abstract

Osteoarthritis (OA) is a prevalent form of arthritis that affects over 32.5 million adults worldwide, causing significant cartilage damage and disability. Unfortunately, there are currently no effective treatments for OA, highlighting the need for novel therapeutic approaches. Thrombomodulin (TM), a glycoprotein expressed by chondrocytes and other cell types, has an unknown role in OA. Here, we investigated the function of TM in chondrocytes and OA using various methods, including recombinant TM (rTM), transgenic mice lacking the TM lectin-like domain (TM^LeD/LeD^), and a microRNA (miRNA) antagomir that increased TM expression. Results showed that chondrocyte-expressed TM and soluble TM [sTM, like recombinant TM domain 1 to 3 (rTMD123)] enhanced cell growth and migration, blocked interleukin-1β (IL-1β)-mediated signaling and protected against knee function and bone integrity loss in an anterior cruciate ligament transection (ACLT)-induced mouse model of OA. Conversely, TM^LeD/LeD^ mice exhibited accelerated knee function loss, while treatment with rTMD123 protected against cartilage loss even one-week post-surgery. The administration of an miRNA antagomir (miR-up-TM) also increased TM expression and protected against cartilage damage in the OA model. These findings suggested that chondrocyte TM plays a crucial role in counteracting OA, and miR-up-TM may represent a promising therapeutic approach to protect against cartilage-related disorders.

## 1. Introduction

Osteoarthritis (OA), the most common form of joint disease, affected approximately 303 million people globally in 2017 [1]. The major feature of OA is the destruction of articular cartilage, often accompanied by synovial inflammation, joint capsule hypertrophy, osteophyte formation, and subchondral bone thickening [2]. The pathogenesis of OA is complex and multifactorial. Chondrocytes play a central role in controlling articular cartilage structure and function, thus regulating the turnover of extracellular matrix components, including collagen, glycoproteins, proteoglycans, and hyaluronan, in maintaining tissue homeostasis [3]. Despite increased knowledge of the function of chondrocytes and the underlying pathology associated with OA and chondro-cartilage disorders, disease-improving treatments and preventative strategies for patients with OA are lacking. Therefore, novel, affordable, and practical treatment approaches for OA are urgently required.

Thrombomodulin (TM), known as CD141, is a type-I transmembrane glycoprotein expressed by several cell types, including chondrocytes, osteoblasts, endothelial cells, monocytes/macrophages, and keratinocytes [4,5,6,7,8]. TM has five structural domains from the N- to the C-terminus: domain 1 (TMD1), a C-type lectin-like domain; domain 2 (TMD2), containing six epidermal growth factor (EGF)-like structures; domain 3 (TMD3), a serine/threonine-rich domain; domain 4 (TMD4), the transmembrane domain; and domain 5 (TMD5), the cytoplasmic domain [9]. These different domains of TM exhibit distinct properties and thus participate in several processes, including protecting against excess inflammation, coagulation, and fibrinolysis; reducing bone loss; and promoting bone repair, cutaneous wound healing, cell proliferation, and cell–cell adhesion [10,11,12,13,14,15,16,17,18]. In addition to being an integral membrane protein, soluble forms of TM (sTM) exist, which comprise various extracellular domains. sTM may be generated in part by cleavage by rhomboid-like-2 membrane protease (RHBDL2) [19].

Cell surface functional expression of TM is regulated via transcriptional and post-transcriptional mechanisms [20]. For example, in endothelial cells, TM is increased by the transcription factor, Krüppel-like factor 2 (KLF2) [21], which is upregulated via inhibition of miR-150. Interestingly, inhibition of miR-150 protects chondrogenic cells ATDC5 against cytokine (IL-1)-induced injury [22]. However, previous studies indicate that TM, expressed by cells such as osteoclasts and osteoblasts, inhibits bone loss and promotes bone healing [17,18]. In addition, its specific role in chondrocytes, cartilage homeostasis, and articular cartilage-related disorders remains limited.

In this study, we explored the physiological functions of TM in chondrocytes and OA, aiming to identify TM as a novel therapeutic target. The experiments were conducted using recombinant sTM comprising extracellular domains 1-3 (rTMD123), transgenic mice lacking the lectin-like domain of TM (domain 1) (TM^LeD/LeD^ mice), and an antagonist of miR-150, referred to as miR-up-TM.

## 2. Results

### 2.1. Chondrocyte Exposure to the Rhomboid Protease RHBDL2 Liberates Soluble TM (sTM), Inducing Cell Proliferation and Migration

The levels of soluble TM in the synovial fluid of patients with rheumatoid arthritis are elevated [23]. However, there is currently no direct evidence to support a connection between TM expression and chondrocyte function and integrity. We first tested whether chondrocytes expressed TM or released sTM. The human chondrocyte cell line, TC28a2, was cultured to confluence under serum-free conditions. Cell lysates and conditioned media (CM) were collected daily for 72 h for western blotting to evaluate TM and sTM expression. Under these conditions, chondrocyte TM steadily increased over time (Figure 1A), with significantly increased sTM production detected in CM at 72 h (Figure 1B). When cells were co-incubated with a specific inhibitor of RHBDL2 (DCI), the release of sTM reduced in a dose-dependent manner (Figure 1C). In this study, the sTM contained the TM extracellular domain (domains 1 to 3, TMD123) because the cutting site of RHBDL2 was TMD4 [19].

Induction of chondrocyte proliferation and migration may promote the healing of osteochondral defects [24,25]. We demonstrated that the introduction of exogenous recombinant TMD123 (rTMD123) led to a dose-dependent improvement in cell proliferation and wound recovery in both the TC28a2 cell line (Figure 1D) and primary human articular chondrocytes (NHAC-kn) (Figure 1E). The effects of TM on cell growth and migration were abrogated by the addition of TM-specific shRNA (shTM) (Figure 1H,I), which significantly reduced TM protein levels (Figure 1F) and cell growth (Figure 1G).

In summary, chondrocytes produce full-length and soluble TM forms that may participate in cellular proliferation and migration.

### 2.2. rTMD123 Inhibits STAT3/MMP 13 Signaling and IL-1β-Mediated Suppression of TM Expression in Chondrocytes

Proinflammatory cytokines, including interleukin (IL)-1β, tumor necrosis factor α (TNFα), IL-6, IL-15, IL-17, and IL-18, and IL-6/STAT3/MMP 13 signaling have been implicated in OA progression [26,27]. We tested whether chondrocyte TM is regulated by IL-1β and whether such a relationship may be relevant in chondrocyte function. In vitro studies revealed that IL-1β decreased TM protein levels, sTM release, and cell proliferation (Figure 2A–C). The dampening effects of IL-1β on cell proliferation and migration were abrogated by rTMD123 treatment. Even in the presence of IL-1β, rTMD123 promoted cell proliferation and migration and inhibited IL-1β-enhanced STAT3/MMP 13 signaling (Figure 2D–H). These findings are consistent with the hypothesis that enhanced chondrocyte TM expression and the administration of TMD123 may protect against OA.

### 2.3. rTMD123 Protects Mouse Knees from ACLT-Induced OA and Dysfunction, Increases Articular Cartilage TM and Reduces MMP 13

To evaluate the in vivo effects of TM on chondrocytes and joint function, we used the well-established mouse model of ACLT-induced OA. rTMD123 was injected into the knee joints of the mice once a week after ACLT surgery (Figure 3A). rTMD123 injection significantly counteracted the reduced weight-bearing capacity of ACLT in a dose-dependent manner, with the maximal benefit achieved at four weeks (Figure 3B). Similar beneficial results were evident in the running test (Figure 3C). Immunohistochemical (IHC) staining and western blotting of articular cartilage showed that rTMD123 treatment maintained TM levels and inhibited MMP13 expression levels in the presence of OA (Figure 3D–H). Furthermore, the TM protein level was also dramatically reduced in the articular cartilage sections of patients with OA (Appendix A). Thus, our in vitro and in vivo results both support a potential role for TM and sTM in protecting chondrocytes from the pathological changes associated with OA.

### 2.4. rTMD123 Rescues Knee Dysfunction and Chondro-Cartilage Joint Damage in TM^LeD/LeD^ Mice with OA

In different acute models, inflammatory arthritis develops more rapidly and severely in TM^LeD/LeD^ mice than in wild-type (TM^wt/wt^) mice [28]. There is no significant difference in appearance between TM^LeD/LeD^ mice and TM^wt/wt^ mice. However, we found that weight-bearing and running time tests following ACLT surgery resulted in a significant loss of knee function in TM^LeD/LeD^ mice compared to that in TM^wt/wt^ mice after two weeks (Figure 4B,C). Notably, articular joint injection of rTMD123 protected TM^LeD/LeD^ mice from knee dysfunction while also preventing the loss of articular cartilage (Figure 4D–G). These results further confirm the importance of TM, the lectin-like domain of TM, and TMD123 in models of acute inflammatory arthritis and OA and chondrocyte function and integrity.

### 2.5. rTMD123 Protects against Knee Dysfunction and Articular Cartilage Loss after OA-Injury Induction

To better represent the clinical situation, we assessed the efficacy of administering rTMD123 after joint damage was induced using the ACLT-OA model. When TM injections were initiated one week after ACLT surgery (Figure 5A), protection of joint function and cartilage integrity was still achieved (Figure 5B–D).

### 2.6. TM Silencing Inhibits miR-150 Antagomir (miR-up-TM)-Increased KLF2/TM Expression and Abolishes the TM-Mediated Protective Effects on Knee Dysfunction in the OA Model

A specific miRNA inhibitor enhances KLF2/TM expression in human endothelial cells; however, its effects on chondrocytes or OA remain unclear [21]. In vitro, we first showed that miR-up-TM dose-dependently enhanced the expression of KLF2 and TM in chondrocytes (Figure 6A–C), and this effect was associated with increased cell proliferation and interference with IL-1β-suppressed cell growth (Figure 6E,F). The miR-up-TM-enhanced TM levels and their associated benefits were significantly reduced by shTM (Figure 6D–F). In line with our in vitro findings, the administration of miR-up-TM to mice following ACLT surgery yielded similar protective effects, with reduced knee dysfunction and cartilage loss (Figure 6G,H). These beneficial responses were abolished by shTM, which caused TM silencing and increased MMP 13 (Figure 6G–L).

## 3. Discussion

TM has been best characterized for its protective properties in the cardiovascular system and inflammation [29,30], whereas its role in other biological systems and related cells has been studied to a much lesser extent. In our previous report, we demonstrated that sTM enhances the functions of osteoblasts through the activation of the fibroblast growth factor receptor (FGFR) signaling pathway [17]. In this study, we demonstrated for the first time that TM is present in chondrocytes and sheds its extracellular domains in the form of sTM, at least in part, via cleavage by the intramembranous rhomboid protease RHBDL2. In vitro and in vivo, we further showed that the proinflammatory cytokine (IL-1β) inhibits chondrocyte TM expression, leading to increased STAT3/MMP 13 signaling and OA formation; these effects were notably attenuated by treatment with rTMD123 (Figure 7A). These findings suggest that TM may be a pivotal blocker of IL-1β-suppressed cell growth and cartilage loss in chondrocytes. Therefore, TM and TMD123 are potential novel agents for the treatment and prevention of OA. An additional intriguing approach to modulating TM expression in OA was further revealed by our studies with the miRNA-150 antagonist, which enhanced the expression of KLF2 and TM and interfered with OA progression (Figure 7B).

OA is characterized most prominently and reproducibly by chondrocyte and cartilage degeneration [31,32]. Previous studies have supported interventions that dampen inflammation and augment chondrocyte growth to positively affect cartilage integrity and bone remodeling. Although IL-1β antagonists have undergone clinical trials, they were discontinued in phase II due to concerns about hepatotoxicity. Clinical trials with IL-1 receptor antagonists for OA have been less than satisfactory [33]. More promising attempts have been made in a surgery-induced OA model to enhance epidermal growth factor receptor (EGFR) signaling with nanoparticles conjugated to transforming growth factor-α (an EGFR ligand). By this approach, cartilage degeneration was reduced, and the pain was attenuated [34]. Many previous studies have demonstrated that TMD1 of sTM (TMD123) is a multi-faceted anti-inflammatory factor that inhibits inflammatory responses such as those induced by LPS or IL-1β [14,35], whereas TMD2 (EGF-like repeat) has a cell growth-promoting effect [11]. In this study, we found that rTMD123 inhibited the STAT3/MMP 13 pathway under the negative impact of IL-1β and promoted chondrocyte growth and migration. Therefore, we suggest that TMD123 is a protective factor with multiple effects.

The short half-life of sTM in plasma (20 h) means that rTMD123 needs to be administered frequently to maintain its protective effect, which could result in high treatment costs [36]. In contrast, miRNA drugs are simpler to produce than protein drugs, as they do not require protein expression or purification. Although there are still challenges to overcome, such as determining the best administration route, ensuring in vivo stability, and targeting specific tissues and cells, miRNA-based therapeutics hold great promise. For instance, downregulating miR-150 could be a viable clinical approach to reducing IL-1β-induced inflammatory injury in human chondrocytes, by targeting KLF2 [22]. KLF2 promotes the expression of multiple genes but suppresses NF-κB-dependent genes [37]. Our findings with shTM, in both cell and animal models, confirm that elevated TM expression by KLF2 is a crucial factor in the protective effect of miR-150 antagonists.

Limitations of this study include the difficulty in establishing chondrocyte-specific TM-deleted mice using clustered, regularly interspaced short palindromic repeat (CRISPR) gene editing and tissue-specific knockout mice. In cellular experiments, we attempted to create chondrocytes without the TM gene using the CRISPR technique but failed because the cells with TM deletion also lost their growth ability, demonstrating the importance of TM for chondrocyte growth. Given the critical importance of TM in chondrocyte growth, it is reasonable to explain the difficulty in establishing chondrocyte-specific TM-deleted mice. Therefore, TM^LeD/LeD^ mice, which have been used to investigate inflammatory arthritis [28], are currently the appropriate choice for further study of the association between TM and OA. The results of this study showed that knee function and articular cartilage loss were significantly faster in TM^LeD/LeD^ mice after ACLT than in TM^wt/wt^ mice. Previous studies have suggested that the TM lectin-like domain plays a role in cell–cell interactions [12], implying that loss of TMD1 leads to the destabilization of cartilage structures and thus accelerates cartilage degeneration under negative stress.

In conclusion, our study demonstrated the role of TM in chondrocytes and sTM in OA. Chondrocyte TM mediates cell growth and migration through its extracellular structural domain, contributing to counteracting IL-1β-induced cell death and OA progression; this indicates the therapeutic potential of TM in OA. The miRNA antagonist-enhanced KLF2/TM cascade provides protection against IL-1β-induced loss of function and OA, which may be a reasonable option for the prevention and treatment of OA.

## 4. Materials and Methods

### 4.1. Antibodies and Reagents

Antibodies recognizing human TM (sc-13164), mouse TM (sc-7097), and glutathione S-transferase GST (sc-138) were obtained from Santa Cruz Biotechnology (Santa Cruz, CA, USA). The following antibodies were purchased from Cell Signaling Technology (Danvers, MA, USA): anti-STAT3 (9139) and anti-p-STAT3 (Tyr705, 9145). Antibodies against GAPDH (ab8245) and MMP13 (ab39012 and ab237604) were purchased from Abcam (Cambridge, UK). The recombinant GST protein (ab70456) was purchased from Abcam (Cambridge, MA, USA). The RHBDL2 serine protease inhibitor (3,4-dichloroisocoumarin [DCI]) and IL-1β (H6291) were purchased from Sigma-Aldrich (St. Louis, MO, USA). Lipofectamine-3000, mirVana^®^ miRNA inhibitor (antagomir) of miR-150 (#MH10070), and the corresponding negative control miRNA were purchased from Thermo Fisher Scientific (Waltham, MA, USA).

### 4.2. Expression of Recombinant TM Domains

As previously described [14], the pCR3-EK vectors (Invitrogen, San Diego, CA, USA) were used to express recombinant TM functional domains for purification and detection in human embryonic kidney 293 mammalian protein expression systems. Expressed recombinant proteins were applied to a nickel-chelating Sepharose column (Amersham Pharmacia Biotech, Piscataway, NJ, USA). Next, recombinant TM domain-containing fractions were eluted, and purified fractions were pooled for use. Purified rTMD123 proteins were examined by Coomassie blue staining and western blotting after gel electrophoresis.

### 4.3. Cell Culture

The primary human articular chondrocytes (NHAC-kn) were purchased from Lonza (Walkersville, MD, USA), cells were cultured according to manufacturer’s recommendations. The human articular chondrocyte cell line (TC28a2) was obtained from the American Type Culture Collection (Rockville, MD, USA) and cultured in Dulbecco’s modified Eagle’s medium (DMEM, Gibco, MD, USA) containing 10% fetal bovine serum (FBS, Gibco). Cells were cultured in a humidified atmosphere at 37 °C and 5% CO_2_. Confluent cells were cultured in six-well plates under serum-free conditions. After incubation with RHBDL2 inhibitor (DCI), cell-free conditioned media (CMs) were collected with the addition of glutathione S-transferase (GST; 20 μg/sample) as an internal control and then concentrated using Centricon tubes with a 10 kDa molecular weight cutoff (Amicon, Beverly, MA, USA). The concentrated samples and cell lysates were then separated using SDS-PAGE and subsequently analyzed using western blotting. To generate TC28a2 cells that did not express TM (referred to as TM-silenced cells), the pSM2c vector system (GenDiscovery Biotechnology, New Taipei, Taiwan) expressing short hairpin RNA (shRNA) against TM (shTM) was transfected [38].

### 4.4. Western Blotting

After SDS-PAGE, samples were transferred onto PVDF membranes (MilliporeSigma, Burlington, MA), blocked with 3% BSA-TBST (50 mM Tris-HCl, 150 mM NaCl, Tween-20; MilliporeSigma), and then probed with primary antibodies. Following incubation with the appropriate secondary antibodies, signals were detected using an enhanced chemiluminescence reagent (Amersham Pharmacia Biotech) using the LAS3000 imaging system (Fujifilm, Stamford, CT, USA). ImageJ software (NIH, Bethesda, MD, USA; https://imagej.nih.gov/ij/ (accessed on 25 July 2022)) was used to quantify the band intensities.

### 4.5. Cell Proliferation/Viability Assay

Cells were seeded into 24-well plates (3 × 10^3^ cells/well in 600 μL) and incubated with recombinant TM proteins at 37 °C in a 5% CO_2_ atmosphere. The medium was replaced every two days. Cell proliferation and viability were quantified using an assay kit (WST-1, K301-500, BioVision; MTT, ab232855, Abcam) according to the manufacturer’s instructions, and the absorbance of acetic-acid-stopped reactions was measured at 450 nm or 590 nm (SPECTRAmax 340, Sunnyvale, CA, USA).

### 4.6. In Vitro Wound Healing Assay

In a six-well plate, when the cells reached confluence, a cell-free zone of approximately 0.6–0.8 mm wide was created by scratching the surface using a 10 μL pipette tip. Following thorough washing with medium, the cells were incubated with rTMD123-containing medium at 37 °C for 48 h. Daily microscopic images of the cells were captured using an Olympus microscopy system (Tokyo, Japan) after the initial wound was made.

### 4.7. Transwell Cell Migration Assay

Cell migration was evaluated using a 24-well chemotaxis chamber with a membrane of 8 μm pore size (Transwell; Corning, Corning, NY, USA). A cell suspension (5 × 10^4^ cells/100 μL of serum-free medium) was added to the upper chamber, and rTMD123 in serum-free medium (600 μL) was added to the lower chamber. Thereafter, the chambers were incubated at 37 °C for 12–24 h. Cells that did not migrate were wiped off the membrane using a cotton swab. The filter was developed using Liu’s stain kit (Shih-Yung Medical Instruments, Taipei, Taiwan), and the number of remaining cells were counted by direct visualization under a light microscope.

### 4.8. Animals

Mice (10 weeks old) lacking the TM lectin-like domain (TM^LeD/LeD^, a gift from the author EMC) [35], and corresponding wild-type C57BL/6 mice (Jackson Laboratory, Bar Harbor, ME, USA) were used. The animal care and experimental procedures were approved by the Institutional Animal Care and User Committee of Kaohsiung Medical University, Kaohsiung, Taiwan (Approval no: KMU-IACUC-109089).

### 4.9. Anterior Cruciate Ligament Transection (ACLT)-Induced Knee OA

Male mice (aged 10 weeks, 18–22 g) were randomly divided into two groups: ACLT and ACLT+ rTMD123 groups (*n* = 5/group). A total of about 50 mice were used. Under general anesthesia, both hind limbs were shaved and prepared for surgery under sterile conditions, as previously described [39]. In experimental animals, OA was induced in the right knee, whereas a sham operation was performed on the left knee (single cutaneous incision and stitching). rTMD123 dissolved in phosphate-buffered saline was injected into the right knees of mice once per week for four weeks, after which the mice were sacrificed by anesthesia overdose.

### 4.10. Intra-Articular Injection in OA Mice Model

Articular injections were administered using a micro syringe with a 34 G needle. The indicated rTMD123, a liposome encapsulating miRNA inhibitor, in a final volume of 20 μL, was injected intra-articularly once a week. After sacrifice, the knees were surgically excised and subjected to histological analysis.

### 4.11. Weight-Bearing Distribution Test

The effect of joint damage on weight distribution in the knees of mice was measured using a dual-channel weight averager (Singa Technology, Taipei, Taiwan), which independently quantifies the weight-bearing ability of each hind paw. Changes in hind paw weight distribution between OA and contralateral control limbs were used as an index of joint discomfort in the OA knee. Mice were placed in an angled Plexiglas chamber positioned such that each hind paw rested on a separate force plate. The force exerted by each hind limb was averaged over a 5 s period, and each data point was the mean of 3 readings of 5 s each. The change in hind paw weight distribution was calculated by determining the percentage difference in the weight between the left and right limbs [40]. The weight-bearing tests were performed before ACLT surgery and each subsequent week until the mice were euthanized.

### 4.12. Treadmill Test

Mice were habituated to run on a Columbus Instruments rodent treadmill (Columbus, OH, USA), with training sessions performed before the ACLT surgery for 15 min/day at a speed of 10–15 m/min for 1 week. After the adaptation period, treadmill tests were performed twice weekly, and the data were averaged after treatment. All the mice were evaluated using an exercise program that consisted of a speed of 40 m/min. The recording time for running endurance was limited to 15 min, and running stopped at the maximum duration of running endurance.

### 4.13. Histology and Immunohistochemistry

As previously described [41], the isolated proximal tibiae were fixed in 10% neutral buffered formalin and decalcified in 10% formic acid after euthanasia. Subsequently, 5 μm microsections were prepared in the coronary plane, stained with glycosaminoglycan with safranin O–Fast Green (1% safranin O counterstained with 0.75% hematoxylin and then 1% Fast Green; Sigma) or toluidine, and quantified with Image-Pro Plus 5.0 software (Media Cybernetics, Rockville, MD, USA). The density of the red-stained area relative to the total area (density/total area) in each group was calculated. The results of the histological study were assessed using microscopic scoring as recommended by the Osteoarthritis Research Society International (OARSI).

For immunohistochemical staining, endogenous peroxidase in the tissues was blocked with 3% hydrogen peroxide, and the samples were digested with enzymes for epitope retrieval. Thereafter, the sections were blocked with FBS for 1 h and incubated with primary antibodies against TM and MMP 13 at 37 °C for 4 h. Subsequently, the EXPOSE mouse- and rabbit-specific horseradish peroxidase-diaminobenzidine detection immunohistochemistry kit (Abcam, Cambridge, MA, USA) was used. Finally, the sections were counterstained with hematoxylin. The data were quantified using ImageJ software by defining the immunostaining of positive cells.

### 4.14. Statistical Analysis

Continuous data are expressed as mean ± SD. The Student’s *t*-test or Mann–Whitney U test was used to determine the significance of comparisons between the two groups. One-way ANOVA followed by post hoc analysis (Tukey’s test) was used for comparisons between more than two groups. *p* < 0.05 was considered statistically significant.

## Figures and Tables

**Figure 1 ijms-24-09522-f001:**
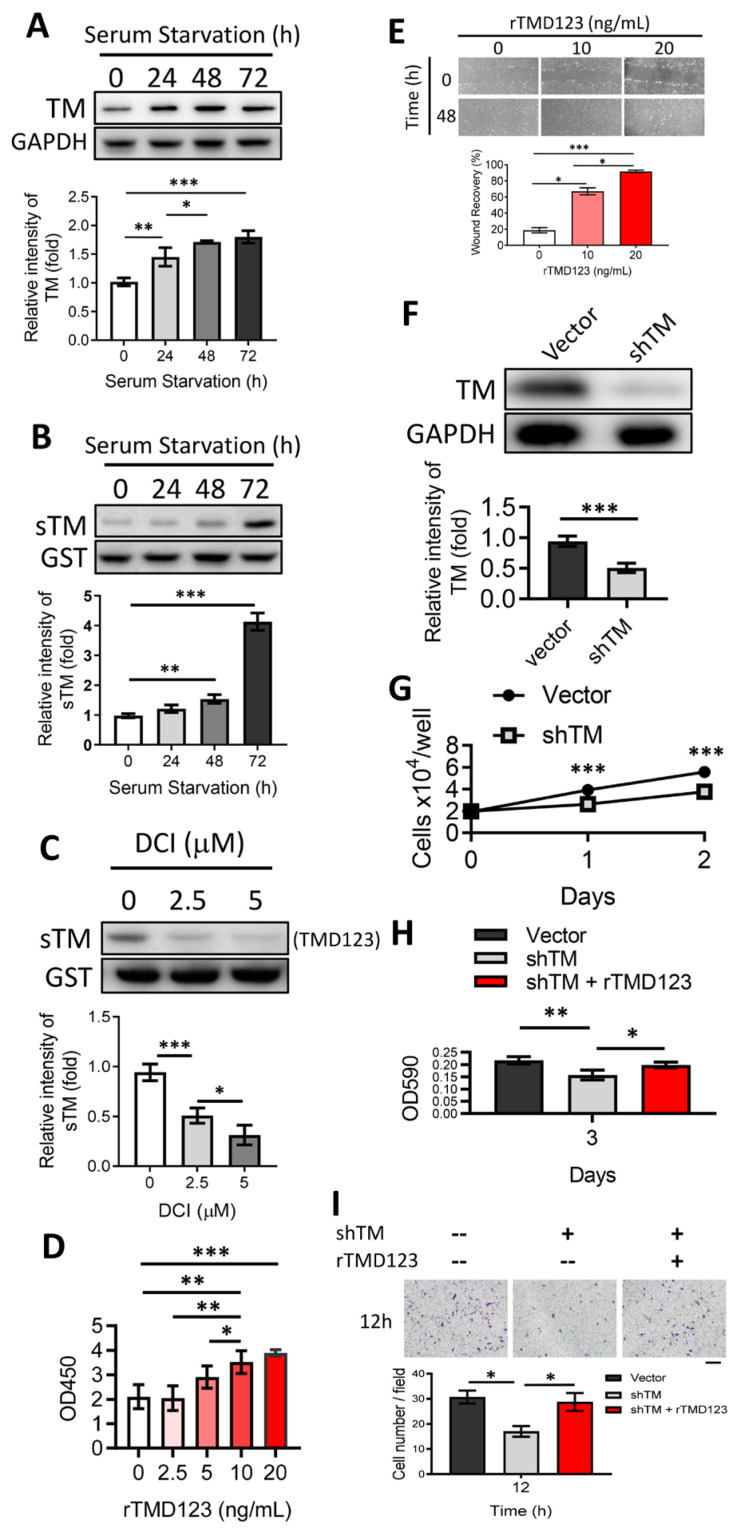
Shedding of the TM extracellular domain by RHBDL2 contributes to cell proliferation and migration in chondrocytes through its EGF-like domain. (**A**) Human chondrocytes (TC28a2) were incubated under serum starvation. Then, the cell lysates were collected at indicated time points for western blotting to evaluate the TM protein level. GAPDH was used as an internal loading control. (**B**) The concentrated conditioned media (CMs) from (**A**) were used to evaluate the protein level of soluble TM (sTM). Glutathione S-transferase (GST) was added as an internal control. (**C**) After incubation for 48 h, the harvested serum-free CMs with DCI (RHBDL2 inhibitor) were used to evaluate the sTM production. (**D**) The WST-1 cell proliferation assay was performed to evaluate the effect of recombinant TMD2-3 (rTMD23; TMD2 is an EGF-like repeat domain) on chondrocytes after treatment for 48 h. (**E**) rTMD123 exhibited a dose-dependent promotion of wound recovery in primary human articular chondrocytes (NHAC-kn). (**F**) TM protein level was significantly reduced by TM shRNA (shTM). (**G**) The growth curve of shTM-transfected TC28a2 cells in two days. (**H**) Effect of rTMD23 on shTM chondrocytes was assessed using MTT cell proliferation assay. (**I**) The transwell cell migration assay was used to evaluate the effect of rTMD23 on shTM chondrocytes. Scale bar: 100 μm. * *p* < 0.05; ** *p* < 0.01; *** *p* < 0.001. All experiments were repeated at least three times.

**Figure 2 ijms-24-09522-f002:**
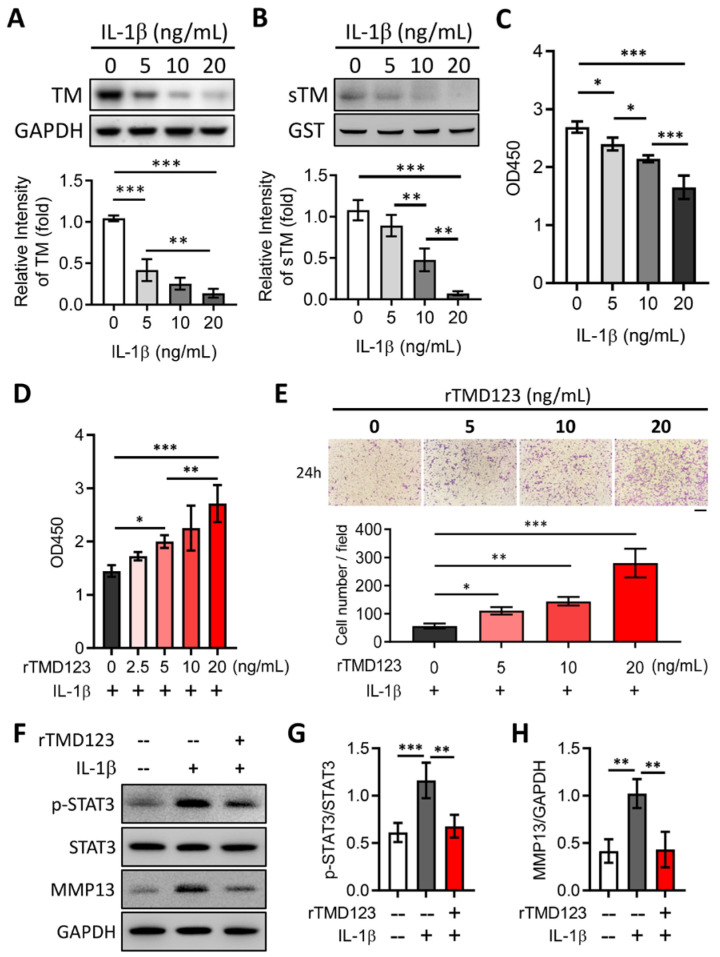
rTMD123 treatment attenuates IL-1β-reduced TM and sTM protein levels, chondrocyte proliferation, and migration by reducing STAT3 signaling and MMP13 expression. Human chondrocytes (TC28a2) were treated with IL-1β for 24 h. Then, the cell lysates and conditioned medium both were collected to evaluate the TM protein level (**A**) and sTM production (**B**). GST was used as an internal control. (**C**) The WST-1 cell proliferation assay was used to evaluate the effect of IL-1β on chondrocytes after treatment for 48 h. (**D**) rTMD123 reversed IL-1β-reduced cell proliferation in a dose-dependent manner. (**E**) The transwell cell migration assay was performed to assess the effect of rTMD123 on IL-1β-inhibited cell migration. Scale bar: 100 μm. (**F**–**H**) IL-1β-mediated STAT3 signaling and MMP13 were arrested by treatment with rTMD123. * *p* < 0.05; ** *p* < 0.01; *** *p* < 0.001.

**Figure 3 ijms-24-09522-f003:**
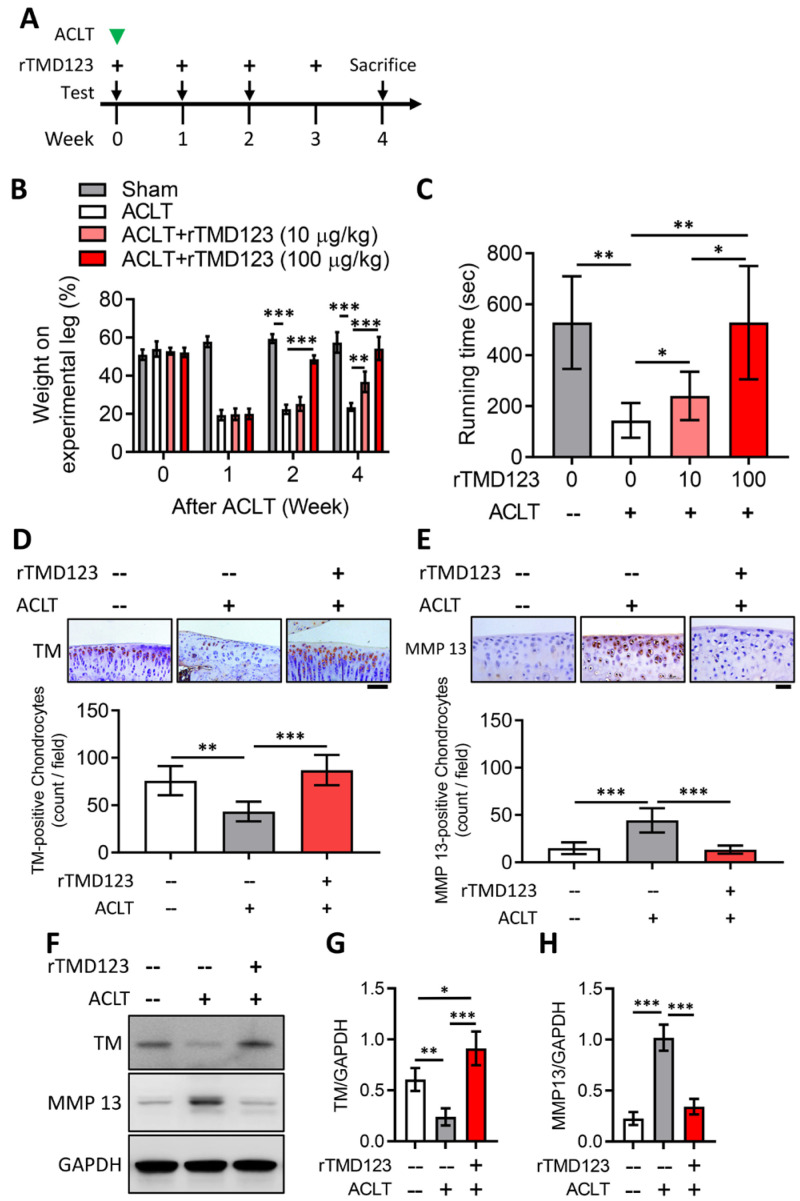
rTMD123 treatment protects knee functions in ACLT-induced OA mice by increasing articular cartilage TM expression and reducing MMP 13 level. (**A**) Illustrated experimental design. The green arrowhead indicates the ACLT surgery was performed. The “+” indicates the injection of rTMD123. The black arrows indicate mice undergoing tests to assess knee function. (**B**) Weight-bearing distribution test. (**C**) Results of treadmill test at week 4. *n* = 5 in each group. Mice were sacrificed at week 4 after the ACLT surgery, and knee joint samples were collected for IHC staining (**D**,**E**) and western blotting analysis (**F**) to evaluate the expression of TM and MMP 13. The red-brown color represents signal-positive cells. (**G**,**H**) Quantitative results of (**F**). Scale bar: 50 μm. * *p* < 0.05; ** *p* < 0.01; *** *p* < 0.001.

**Figure 4 ijms-24-09522-f004:**
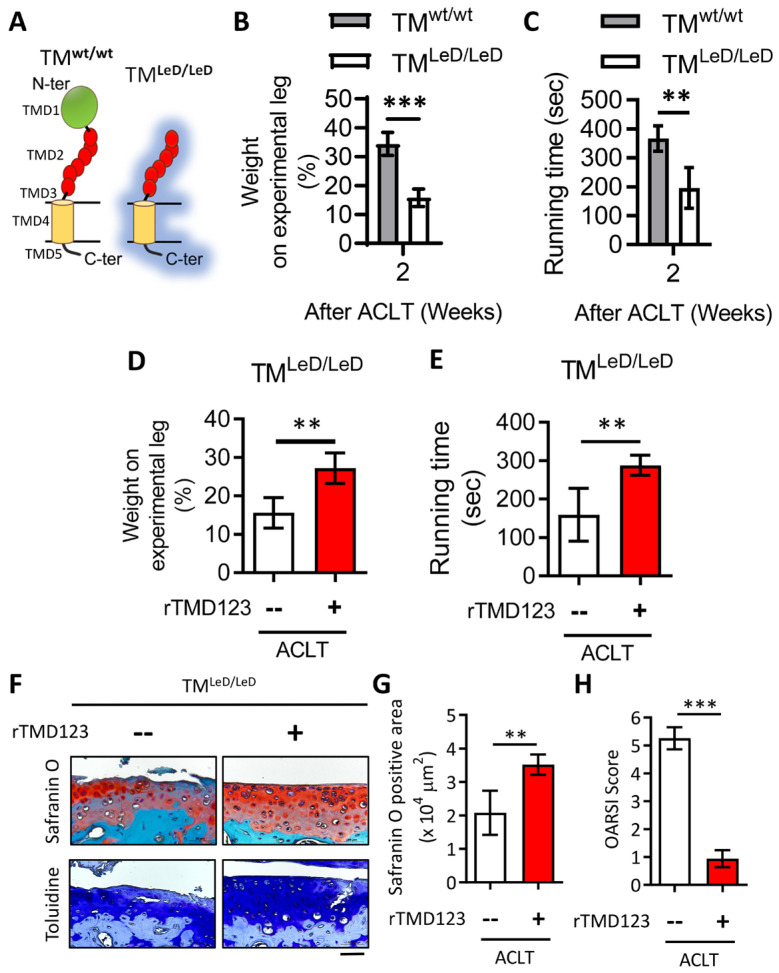
Accelerated knee joint function deterioration due to deletion of TM lectin-like domain after ACLT surgery could be improved by rTMD123. (**A**) Graph showing the difference of TM protein structure between TM^wt/wt^ and TM^LeD/LeD^. (**B**) Weight-bearing distribution test. (**C**) Treadmill test. Results of the weight distribution test (**D**) and treadmill test (**E**) two weeks after ACLT surgery with or without rTMD123 (100 μg/kg) injection. (**F**) After surgery and treatment with rTMD123 for four weeks, knee sections were stained with safranin O and toluidine blue to evaluate the cartilage area (**G**) and OARSI score (**H**). Scale bar: 50 μm. ** *p* < 0.01; *** *p* < 0.001.

**Figure 5 ijms-24-09522-f005:**
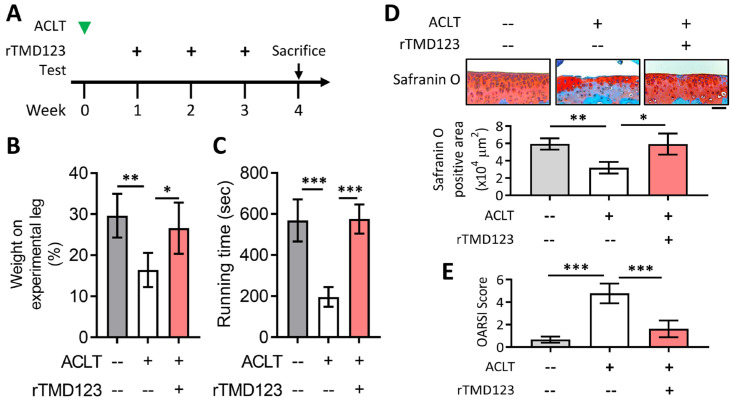
Delayed administration of rTMD123 one week after ACLT surgery still protected knee function in mice. (**A**) Illustrated experimental design. The green arrowhead indicates the ACLT surgery was performed. The “+” indicates the injection of rTMD123. The black arrow indicates mice undergoing tests to assess knee function. (**B**,**C**) Results of weight-bearing distribution test and treadmill test. (**D**) Quantitative results of safranin O staining. (**E**) OARSI score. Scale bar: 50 μm. * *p* < 0.05; ** *p* < 0.01; *** *p* < 0.001.

**Figure 6 ijms-24-09522-f006:**
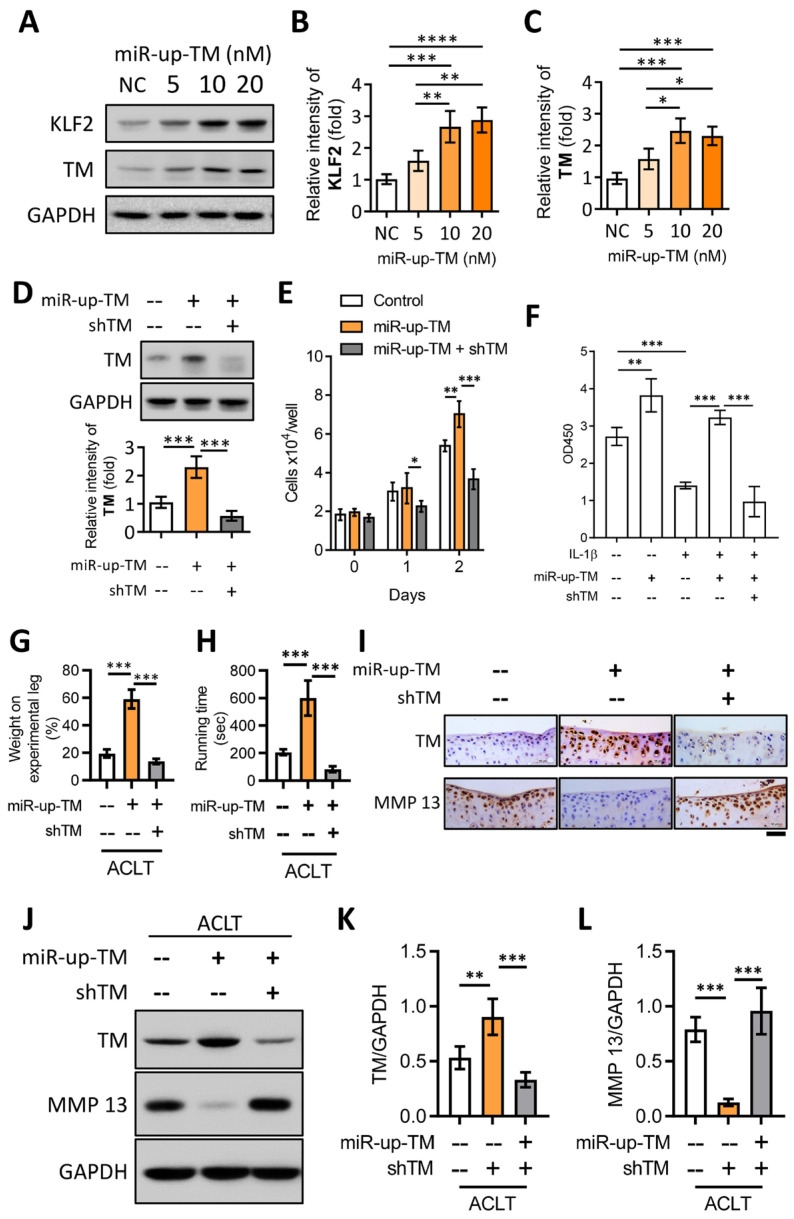
miR-up-TM-enhanced KLF2/TM signaling is critical to protect knee functions by promoting chondrocyte growth and reducing MMP 13 level. (**A**) After transfection with negative control miRNA (NC, 20 nM) and miR-up-TM (5-20 nM) for 24 h, cell lysates of TC28a2 were analyzed using western blotting. (**B**,**C**) Quantified expression levels of KLF2 and TM. (**D**) Western blotting results showed that miR-up-TM-enhanced TM levels were abrogated by shTM. (**E**) shTM inhibited the growth of chondrocytes boosted by miR-up-TM. (**F**) After two days of treatment, the positive benefit of miR-up-TM on IL-1β-suppressed chondrocyte survival was abolished by shTM. (**G**,**H**) Four weeks after ACLT surgery, the protective effect of miR-up-TM on knee functions in TM^wt/wt^ mice was lost due to shTM. (**I**) Staining of knee sections showed that miR-up-TM enhanced articular cartilage expression and inhibited MMP 13 expression; this effect could be suppressed by shTM. (**J**–**L**) The analytical and quantitative results of western blotting show a trend consistent with (**I**). Scale bar: 50 μm. * *p* < 0.05; ** *p* < 0.01; *** *p* < 0.001; **** *p* < 0.0001.

**Figure 7 ijms-24-09522-f007:**
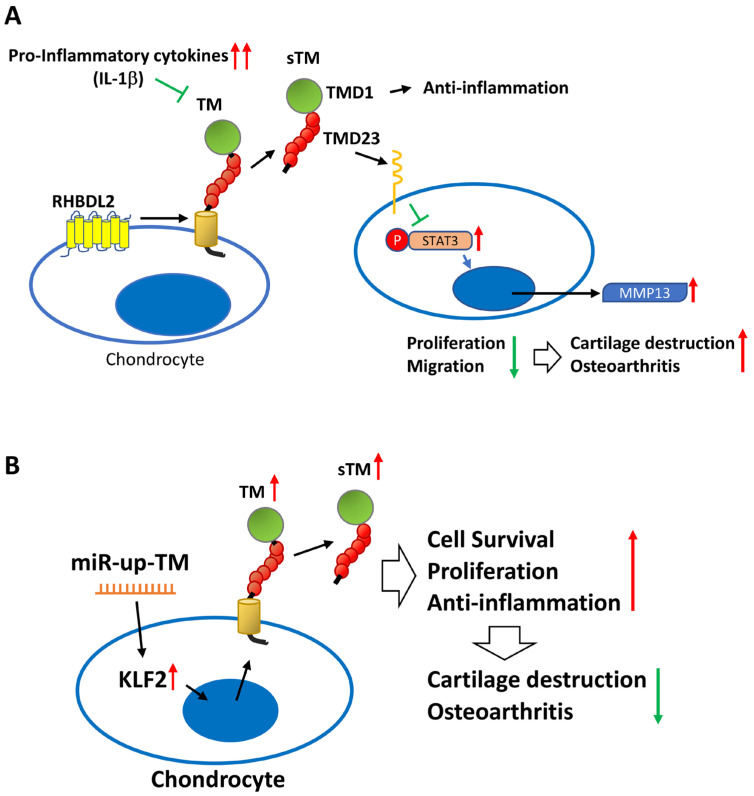
The schematic diagram shows the role of the TM in chondrocytes and OA. (**A**) Excessive inflammatory factors (e.g., IL-1β) impede chondrocyte TM performance and RHBDL2-mediated sTM production, leading to elevated STAT3/MMP 13 messaging pathways and reduced cell growth and migration, ultimately contributing to articular cartilage damage and OA. (**B**) miR-up-TM (miR-150 antagomir) enhances TM expression and sTM release from chondrocytes by increasing KLF2 transcription factors, which in turn improves the anti-inflammatory and growth capacity of the cells, eventually leading to anti-OA effects.

## Data Availability

The original data of this present study are available from the corresponding authors.

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
