# Peer review of "Chondrocyte Thrombomodulin Protects against Osteoarthritis"

_ijms, 2023, doi:10.3390/ijms24119522_

Round 1
Reviewer 1 Report
This is a well-organized and nicely written article describing that thrombomodulin (TM) protects deterioration of articular cartilage using mouse models of OA, ACLT-OA and TMLeD/LeD mice. These in vivo studies providing enough information of the function of TM on OA.
The authors also described the protective mechanism of TM on OA by in vitro studies using a human articular chondrocyte cell line, TC28a2. The in vitro study which is appeared before the in vivo study nicely describes the biological significance of TM on articular chondrocytes. However, I think that in vitro study is better to be performed at least in part using primary human articular chondrocytes which are commercially provided.
The authors provided immunohistological findings of the expression of TM using histological pictures of the normal cartilage and that of OA in Supplemental Figure 1. The authors need to show immunohistochemical findings using human histopathological specimens to confirm that deteriorated lesion has less TM than normal portion. This is a very important point for discussing therapeutic application of TM to human OA patients.
Author Response
This is a well-organized and nicely written article describing that thrombomodulin (TM) protects deterioration of articular cartilage using mouse models of OA, ACLT-OA and TMLeD/LeD mice. These in vivo studies providing enough information of the function of TM on OA.
Re: Thank you so much for your positive feedback.
The authors also described the protective mechanism of TM on OA by in vitro studies using a human articular chondrocyte cell line, TC28a2. The in vitro study which is appeared before the in vivo study nicely describes the biological significance of TM on articular chondrocytes. However, I think that in vitro study is better to be performed at least in part using primary human articular chondrocytes which are commercially provided.
Re: Thank you for your comment. Using primary human articular chondrocytes (NHAC-kn) and conducting a scratch-wound healing assay, we observed that rTMD123 exhibited a dose-dependent promotion of cell migration. (Please see Fig. 1E )
The authors provided immunohistological findings of the expression of TM using histological pictures of the normal cartilage and that of OA in Supplemental Figure 1. The authors need to show immunohistochemical findings using human histopathological specimens to confirm that deteriorated lesion has less TM than normal portion. This is a very important point for discussing therapeutic application of TM to human OA patients.
Re: Thank you for your comment. In fact, the IHC results presented in Supplemental Figure 1 depict sections from the knees of both normal human subjects and OA patients. These results clearly demonstrate that the expression of TM in the OA group was nearly undetectable.

Reviewer 2 Report
In this study, the authors investigated the effects of thromobomodulin on cultured chrondrocytes and on experimentally induced osteoarthritis using a murine model. They found that chrondrocytes expressed thrombomodulin and that addition of recombinant thrombomodulin to cultured chondrocytes enhanced their growth and migration. They also found that thrombomodulin inhibited the proinflammatory activities of IL-1beta, and improved the outcome of experimentally induced osteoarthritis. The study is well designed and the outcome of their experiments support their conclusion that thrombomodulin has a potential beneficial role in the treatment of osteoarthritis
Overall, an excellent use of the English language
Author Response
In this study, the authors investigated the effects of thromobomodulin on cultured chrondrocytes and on experimentally induced osteoarthritis using a murine model. They found that chrondrocytes expressed thrombomodulin and that addition of recombinant thrombomodulin to cultured chondrocytes enhanced their growth and migration. They also found that thrombomodulin inhibited the proinflammatory activities of IL-1beta, and improved the outcome of experimentally induced osteoarthritis. The study is well designed and the outcome of their experiments support their conclusion that thrombomodulin has a potential beneficial role in the treatment of osteoarthritis.
Re: Thank you so much for your positive feedback.
